# Retroviral Replicating Vectors Mediated Prodrug Activator Gene Therapy in a Gastric Cancer Model

**DOI:** 10.3390/ijms241914823

**Published:** 2023-10-02

**Authors:** Hiroaki Fujino, Emiko Sonoda-Fukuda, Lisa Isoda, Ayane Kawabe, Toru Takarada, Noriyuki Kasahara, Shuji Kubo

**Affiliations:** 1Laboratory of Molecular and Genetic Therapeutics, Institute of Advanced Medical Science, Hyogo Medical University, Hyogo 663-8501, Japanisoda.lisa@gmail.com (L.I.); takarada@kobepharma-u.ac.jp (T.T.); 2Departments of Biomedical Chemistry, School of Science and Technology, Kwansei Gakuin University, Hyogo 669-1330, Japan; 3Laboratory of Functional Molecular Chemistry, Kobe Pharmaceutical University, Hyogo 658-8558, Japan; 4Departments of Neurological Surgery and Radiation Oncology, University of California, San Francisco, CA 94143, USA; noriyuki.kasahara@ucsf.edu

**Keywords:** gastric cancer, retroviral replicating vectors, prodrug activator gene therapy

## Abstract

Retroviral replicating vectors (RRVs) selectively replicate and can specifically introduce prodrug-activating genes into tumor cells, whereby subsequent prodrug administration induces the death of the infected tumor cells. We assessed the ability of two distinct RRVs generated from amphotropic murine leukemia virus (AMLV) and gibbon ape leukemia virus (GALV), which infect cells via type-III sodium-dependent phosphate transporters, PiT-2 and PiT-1, respectively, to infect human gastric cancer (GC) cells. A quantitative RT-PCR showed that all tested GC cell lines had higher expression levels of PiT-2 than PiT-1. Accordingly, AMLV, encoding a green fluorescent protein gene, infected and replicated more efficiently than GALV in most GC cell lines, whereas both RRVs had a low infection rate in human fibroblasts. RRV encoding a cytosine deaminase prodrug activator gene, which converts the prodrug 5-flucytosine (5-FC) to the active drug 5-fluorouracil, showed that AMLV promoted superior 5-FC-induced cytotoxicity compared with GALV, which correlated with the viral receptor expression level and viral spread. In MKN-74 subcutaneous xenograft models, AMLV had significant antitumor effects compared with GALV. Furthermore, in the MKN-74 recurrent tumor model in which 5-FC was discontinued, the resumption of 5-FC administration reduced the tumor volume. Thus, RRV-mediated prodrug activator gene therapy might be beneficial for treating human GC.

## 1. Introduction

Gastric cancer (GC) is one of the most common malignancies worldwide, with over one million new cases reported in 2020 and an estimated 770,000 deaths, ranking fifth for incidence and fourth for global mortality [1]. Currently, primary therapeutic approaches for GC include surgical resection, chemotherapy, and radiation therapy. Among them, surgical resection is still the preferred approach and may offer a 5-year overall survival of 84% to 96% with endoscopic resection and 98% with gastrectomy [2,3]. However, the 5-year overall survival of patients with locally advanced, unresectable or metastatic GC is 9.2% and the median overall survival is approximately 1 year for patients treated with combination chemotherapy [2]. Thus, GC remains a serious malignancy, and therefore new, effective treatments are required. As a novel treatment for GC, cancer virotherapy has recently gained increasing attention [3,4]. Previously, various oncolytic viruses that directly lyse tumor cells have been developed and evaluated, including herpes simplex virus type 1 [5], adenovirus [6], echovirus 1 [7], Newcastle disease virus [8], reovirus [9], and vaccinia virus [10]. Although these viruses have shown promising results in preclinical studies, there are still challenges for their translation to effective clinical treatments. 

Retroviral replicating vectors (RRVs), developed based on the amphotropic murine leukemia virus (AMLV), possess unique characteristics that allow highly efficient and tumor-selective gene transfer [11]. The tumor specificity of RRV is ensured by the absolute requirement of cell division for active infection and virus-selective advantages in the tumor microenvironment related to blunted innate immune responses (e.g., APOBECs, DDX41, tetherin, TRIM-5α) [12,13,14,15], as well as suppressed acquired immune responses relative to normal cells [16,17,18]. In contrast to oncolytic viruses, RRVs are not cytotoxic. However, RRVs containing prodrug activator genes such as yeast cytosine deaminase (CD) can induce cell death synchronously by administering a prodrug 5-fluorocytosine (5-FC), which is converted to the anticancer drug 5-fluorouracil (5-FU) [19,20]. This strategy might result in fewer side effects because 5-FU has a very short half-life and is generated only in RRV-infected cells. Moreover, for safety, unintentional RRV propagation is avoided by the administration of prodrugs [11]. Indeed, RRV-mediated prodrug activator gene therapy was effective against a variety of tumors including glioma [21], mesothelioma [22], lung cancer [23], pancreatic cancer [24], and osteosarcoma [25]. Based on these encouraging preclinical results, clinical trials of RRV-mediated prodrug activator gene therapy have been initiated. Toca 511 (vocimagene amiretrorepvec), an enhanced AMLV-based RRV encoding an optimized CD for therapeutic use [26], has shown highly promising results in Phase I dose escalation multi-center trials for high-grade glioma [27,28], including complete responses and survival benefits. Even though a multicenter, randomized, open-label phase II/III trial of patients with recurrent high-grade glioma failed to meet the study endpoints, the clinical evaluation of Toca 511/5-FC treatment revealed promising therapeutic activity in particular subgroups of glioblastoma patients [11,29], and additional clinical research is currently being conducted. However, the therapeutic effectiveness of RRV in GC cells has not been examined.

We recently developed a new RRV derived from the gibbon ape leukemia virus (GALV) [22]. GALV and AMLV are gammaretroviruses that infect cells through distinct forms of mammalian type-III sodium-dependent phosphate transporters (PiT) [30,31,32]. Specifically, GALV utilizes PiT-1 (SLC20A1), whereas AMLV uses PiT-2 (SLC20A2). As a result, both RRVs have distinct host ranges and do not belong to the same interference class [30,31,32]. Although PiT-1 and PiT-2 are ubiquitously expressed proteins that regulate the intracellular inorganic phosphate balance in normal cells, we previously found that they are expressed differently in various cancer cell lines [22,25,33]. In cells with low receptor expression, RRVs requiring the receptor for viral entry demonstrated restricted replicative spread, and the efficiency of cell death following prodrug administration correlated with the RRV replication kinetics [22,25]. Based on these results, it will be possible to select and administer an RRV that will be effective prior to the start of treatment by examining the expression levels of PiT-1 and PiT-2 receptors in the tumor.

In the present study, we examined prodrug activator gene therapy for experimental human GC using RRVs generated from AMLV and GALV. Our findings suggest that prodrug activator gene therapy with AMLV may be useful for treating GC when the tumor is not amenable to GALV.

## 2. Results

### 2.1. Relative Evaluation of PiT-2 or PiT-1 Expression Levels

We examined the expression levels of cellular receptors for RRVs in human GC cell lines (HGC-27, KATO-III, MKN-7, MKN-45, MKN-74) using q-PCR (Figure 1). The expression levels of PiT-1 (GALV receptor) and PiT-2 (AMLV receptor) in human dermal fibroblasts were similar to those of human embryonic kidney 293 cells as a positive control. Of the GC cell lines, MKN-7 and MKN-74 expressed high levels of PiT-2 compared with PiT-1 expression. Indeed, compared with PiT-1 expression, PiT-2 expression was 17.3 times higher in MKN-7 and 10.6 times higher in MKN-74. Therefore, AMLV may have a propagation advantage over GALV in GC cells.

### 2.2. AMLV Vectors Replicate Efficiently in Human GC Cell Lines In Vitro

To analyze the replication and spread efficiency of AMLV and GALV in culture, human GC cell lines and human fibroblasts were inoculated with AMLV-GFP or GALV-GFP vectors at low multiplicities of infection (MOI) and thereafter analyzed for GFP expression using flow cytometry (Figure 2). GFP-positive cells were considered to be AMLV- or GALV-infected cells. In normal fibroblasts, AMLV and GALV showed no viral spread until day 42 after RRV infection. However, AMLV replicated and spread more efficiently than GALV in all GC cells. In particular, KATO-III, MKN-7, and MKN-74 had low GALV spread, whereas AMLV spread efficiently, with GFP-positive cell rates of more than 90% at 31 days (KATO-III), 17 days (MKN-7), and 28 days (MKN-74) after infection (Figure 2). These findings imply that AMLV has superior viral spread to GALV in cultures of GC cell lines and that its efficiency depends on the expression level of RRV receptors in GC cells (Figure 1).

### 2.3. Prodrug Activator Gene-Mediated Cell-Killing Effect of RRVs on Human GC Cells

We evaluated the cell-killing effect of AMLV-CD and/or GALV-CD combined with 5-FC on GC cells using an AlamarBlue assay (Figure 3). In fibroblasts, AMLV and GALV showed no cell-killing effect at 5-FC 1 mM. However, the viability of GC cell lines (HGC-27, KATO-III, MKN-7, MKN-45, MKN-74) was decreased in a concentration-dependent manner with 5-FC. For all GC cell lines, viability after the addition of 5-FC was significantly lower for AMLV than GALV. For example, the difference in cell viability between both RRVs with 1 mM 5-FC was greater than 30% for HGC-27 and MKN-45 and greater than 40% for KATO-III, MKN-7, and MKN-74. These results indicate that AMLV was more effective at killing GC cell lines compared with GALV. It also suggested that the cytotoxic effect in GC cell lines depended on the efficiency of RRV spread (Figure 2), as well as the expression level of RRV receptors (Figure 1).

### 2.4. AMLV-Mediated CD/5-FC Prodrug Activator Gene Therapy Has Potent In Vivo Antitumor Effects in Primary and Recurrent Subcutaneous MKN-74 Human GC Xenograft Models

To analyze the antitumor effect of RRV-mediated CD/5-FC prodrug activator gene therapy, we administered AMLV-CD, GALV-CD, or PBS intratumorally to MKN-74 subcutaneous tumors in nude mice (Figure 4a). Tumors in the PBS group continued to grow after the start of 5-FC administration. In the AMLV and GALV groups, the tumor volume began to decrease on day 3 after 5-FC administration. However, the GALV-treated group showed tumor shrinkage up to day 10 after 5-FC treatment, although tumor growth was observed thereafter. However, the AMLV-treated group continued to show high antitumor efficacy during the 5-FC administration. After day 17 of 5-FC treatment, the AMLV group had a significantly higher tumor shrinkage rate than the GALV group. These results indicate that the combination of AMLV-CD/5-FC had a superior antitumor effect than GALV-CD/5-FC in a human subcutaneous GC tumor model. This result was also consistent with the superior killing effect of AMLV (Figure 3) on GC cell lines by RRV-CD/5-FC.

Finally, we examined whether the re-administration of 5-FC was effective on recurrent tumors. In the MKN-74 recurrent tumor model (*n* = 6), the re-administration of 5-FC resulted in significant tumor reduction (Figure 4b). Compared with the maximum tumor volume of 2141 ± 646 mm^3^ (day 7), there was a significant decrease of 513 ± 115 mm^3^ (* *p* = 0.022) at day 28 and 557 ± 132 mm^3^ (* *p* = 0.026) at day 31. These results suggest that tumors that have regrown after RRV-mediated prodrug activator gene therapy can be treated again through the re-administration of 5-FC alone, without the re-administration of RRV.

## 3. Discussion

In this study, we demonstrated that compared with GALV, AMLV infection was more efficient in GC cell lines with high PiT-2 receptor expression and had a higher cell-killing effect on GC cells and in subcutaneous GC mouse models. This is the first preclinical study to demonstrate that RRV-mediated prodrug activator gene therapy is a promising novel treatment for GC.

The propagation efficiency, cell-killing effect, and antitumor effect of RRV were higher for GC cells with high RRV receptor expression, suggesting that RRV receptor expression in tumor cells has a significant impact on therapeutic efficacy. We previously reported that the replicative spread of RRVs in solid tumors was affected by the cellular receptor expression levels of human malignant mesothelioma cells [22] and osteosarcoma cells [25]. Interestingly, GALV was superior to AMLV (PiT-1 > PiT-2) in mesothelioma, whereas AMLV was superior to GALV (PiT-1 < PiT-2) in osteosarcoma, as in GC, although the biological relevance of the differential expression between PiT-1 and PiT-2 in these tumor cells is unclear. It is expected that, for clinical applications, it will be possible to select and administer an RRV that will be effective prior to the start of treatment by examining the expression levels of PiT-1 and PiT-2 receptors in a biopsy of the patient’s tumor as markers for personalized cancer virotherapy. Furthermore, other factors might affect the replicative spread of RRVs in solid tumors, such as the cell proliferation rate, degree of remaining innate immunity [34,35] and acquired immune suppression [36,37], as well as receptor expression levels. Therefore, in the future, it will be necessary to collect and culture live tumor cells as clinical specimens to examine RRV propagation.

Although complete tumor shrinkage is desired for cancer therapy, cancer recurrence is common, including in stomach cancer [38]. In this study, local tumor recurrence was found in 9 of 12 mice in the AMLV-CD/5-FC treated group after the 5-FC administration was completed. Of these nine mice, six were restarted on the 5-FC prodrug and had significant tumor volume reduction (Figure 4b). This finding was predicted because RRVs preferentially infect cancer cells and insert viral sequences into chromosomes, resulting in infected tumor cells and the retention of RRVs that continue to express CD in daughter cells. Therefore, as long as infected tumor cells are present, RRV propagation continues within the tumor, leading to long-term tumor growth suppression with prodrug administration. Thus, a major benefit of this therapy is its capacity to restrict tumor growth over time through prodrug administration, without the need to re-administer the virus in the event of tumor recurrence. To reproduce this in future clinical settings, prodrug administration methods will need to be optimized for dose, period, and interval.

To enhance the antitumor effect, a combination therapy with other drugs may be effective. A candidate for combination chemotherapy to treat GC is 5-FU plus tipiracil hydrochloride, which has a therapeutic effect on GC cells resistant to 5-FU [39]. The combination with tipiracil hydrochloride may restore and enhance the effect of 5-FU produced in RRV-infected tumor cells. Another possible combination is to co-infect RRVs that express additional prodrug activator genes. For example, when herpes simplex virus thymidine kinase (HSV-tk) is used, the administration of ganciclovir as a prodrug converts the drug into an anticancer drug that is phosphorylated by HSV-tk in RRV-infected tumor cells, which has a high cytotoxic effect [40]. We previously reported that AMLV and GALV could coinfect and replicate independently in cultured cells without inducing superinfection resistance. This distinguishing characteristic enables dual RRV-mediated CD/5-FC and HSV-tk/ganciclovir combinatorial gene therapy [33]. This combinatorial virotherapy demonstrated synergistic cytotoxic effectiveness compared with single-vector gene treatment. Thus, the co-infection of cancer cells with AMLV and GALV vectors with separate prodrug activator genes may be used for combination intracellular treatment, resulting in increased cytotoxicity while avoiding drug resistance.

In conclusion, our results demonstrate that AMLV vectors can efficiently replicate and achieve effective tumor transduction in human GC cells. This is the first preclinical study to show that RRV vectors efficiently transduce human GC cells and have a therapeutic effect. Therefore, this therapy could be expected to be a new treatment modality for human GC.

## 4. Materials and Methods

### 4.1. Cell Culture

Human dermal fibroblasts (Cambrex Bio Science Walkersville, Walkersville, MD, USA) and human GC cell lines (HGC-27, KATO-III, MKN-7, MKN-45, MKN-74; RIKEN BioResource Center, Ibaraki, Japan, Appendix A) were grown in Roswell Park Memorial Institute 1640 medium (RPMI 1640; Nacalai Tesque, Kyoto, Japan) supplemented with 10% fetal bovine serum (FBS) (Atlas Biologicals, Fort Collins, CO, USA). The human embryonic kidney 293 cell line (Microbix Biosystems, Mississauga, ON, Canada) [41] and transformed human embryonic kidney 293T cell lines [42] were cultured in Dulbecco’s modified Eagle’s medium (DMEM; Nacalai Tesque), supplemented with 10% FBS. All cells were maintained in a humidified incubator at 37 °C with 5% CO_2_.

### 4.2. Plasmid and RRV Production

The AMLV and GALV vector plasmids were generated in a previous study [22]. Briefly, the plasmids AMLV-GFP and GALV-GFP encode AMLV or GALV, carrying an internal ribosome entry site and a GFP coding sequence (IRES-GFP). The plasmids AMLV-CD and GALV-CD were constructed from the above plasmids by replacing the GFP gene with a CD prodrug-activator gene.

For RRV production, 293T cells were transfected with each plasmid using Lipofectamine 2000 (Life Technologies Japan, Tokyo, Japan). At 48 h post-transfection, the virus-containing supernatant was collected, filtered, and stored at −80 °C in aliquots until future use [22]. The titers of RRVs were determined by GFP expression using a FACS Calibur flow cytometer (Becton Dickinson Japan, Tokyo, Japan) and expressed as transduction units (TU) per mL.

### 4.3. Quantitative Real-Time PCR Analysis

Total RNA was isolated from GC cell lines (HGC-27, KATO-III, MKN-7, MKN-45, MKN-74) using Sepasol-RNA I Super G (Nacalai Tesque) according to the manufacturer’s instructions. The concentration of RNA was measured with a NanoDrop spectrophotometer (Thermo Fisher Scientific, Waltham, MA, USA) and adjusted to 20 ng/µL with nuclease-free water after DNase treatment. The TaqMan probe and primers for PiT-1 (Hs00965587_m1), PiT-2 (Hs00198840_m1), and glyceraldehyde-3-phosphate dehydrogenase (GAPDH) (Hs99999905_m1) were purchased from Applied Biosystems Japan (Tokyo, Japan). Quantitative real-time PCR analysis was performed in triplicate using a TaqMan One-Step RT-PCR Master Mix Reagents kit (Applied Biosystems Japan), as described previously [22].

### 4.4. Flow Cytometry of RRV Replication Kinetics

Fibroblasts, HGC-27, KATO-III, MKN-45, and MKN-74 at 10% confluency were infected with AMLV-GFP or GALV-GFP at an MOI of 0.01 in 6-well plates. MKN-7 (low growth rate) at 30% confluency was infected with AMLV-GFP and GALV-GFP at an MOI of 0.01 in a 60 mm dish. At various time points, the cells were rinsed with PBS and harvested via trypsinization. The detached cells were resuspended in PBS and immediately analyzed using flow cytometry on an FACS Calibur flow cytometer. At least 10,000 cells were analyzed per sample.

### 4.5. AlamarBlue Assay

Human Fibroblasts, HGC-27, KATO-III, MKN-45, and MKN-74 were infected with AMLV-CD or GALV-CD at an MOI of 0.01. The cells were seeded in 96-well plates in triplicate wells after 15 days of viral infection. The next day, 5-FC (Wako Pure Chemical Industries, Osaka, Japan) was added into the culture media at a final concentration of 0.1, 1, or 10 mM. After 6 days of 5-FC treatment, the culture media was decanted from the 96-well plates and 40 µL/well of the working solution of AlamarBlue reagent (Alamar Biosciences, Sacramento, CA, USA) was applied to each well. The plates were then incubated for 3 h at 37 °C under 5% CO_2_. Fluorescence was detected at an excitation wavelength of 544 nm and emission wavelength of 590 nm on an Infinite M200PRO multilabel plate reader (Tecan Japan, Kanagawa, Japan).

### 4.6. Subcutaneous Xenograft Model of Human GC

Five-week-old female BALB/c-nu/nu (nude) mice were purchased from The Jackson Laboratory Japan (Yokohama, Japan) and bred under specific pathogen-free conditions at the Hyogo Medical University. To establish a human GC xenograft model, mice were subcutaneously inoculated with 1 × 10^6^ MKN-74. When tumors reached a diameter of 5 mm, mice were randomized into three groups and a volume equivalent to 2 × 10^4^ TU (100 µL) of AMLV-CD (*n* = 13), GALV-CD (*n* = 12), or PBS (*n* = 12) was injected directly into the tumors (day 0).

Mice were given 500 mg/kg/day of 5-FC intraperitoneally three times a week for 35 days starting on day 14 (until day 49). The condition of each mouse was checked daily during this study. The major and minor axes of each tumor were measured with calipers twice a week. The tumor volume was calculated as V = 0.5 × (major axis) × (minor axis)^2^.

An MKN-74 recurrent tumor model was developed after the discontinuation of 5-FC. The 5-FC treatment of AMLV-CD/5-FC-treated mice (*n* = 12) was discontinued and they were observed for 28 days for tumor regrowth. As a result, three mice remained tumor-free, whereas the remaining nine mice showed tumor regrowth. Of the nine mice, six with a similar tumor size were selected for the experiment. When tumor volumes reached ~1000 mm^3^ in the six mice, 5-FC was readministered three times a week for 32 days (day 77 to day 108). The condition of each mouse and the tumor volume were evaluated as described above.

### 4.7. Statistical Analysis

Data are presented as the mean ± SE. All data were analyzed using the Student’s *t*-test. A *p*-value < 0.05 was considered statistically significant.

### 4.8. Abbreviations

All abbreviations are listed in Appendix A.

## Figures and Tables

**Figure 1 ijms-24-14823-f001:**
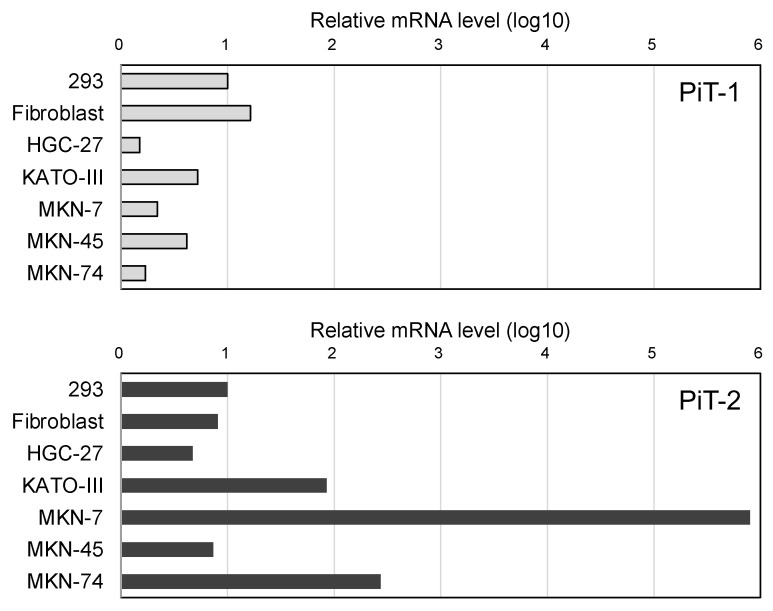
Relative mRNA levels of cellular receptors for RRVs in human gastric cancer cell lines measured with qPCR. The expression levels of PiT-1 (receptor for GALV) and PiT-2 (receptor for AMLV) were compared between human gastric cancer cell lines (HGC-27, KATO-III, MKN-7, MKN-45, MKN-74) and normal cells (fibroblasts, 293). GAPDH was used as an endogenous control for the relative evaluation of expression levels.

**Figure 2 ijms-24-14823-f002:**
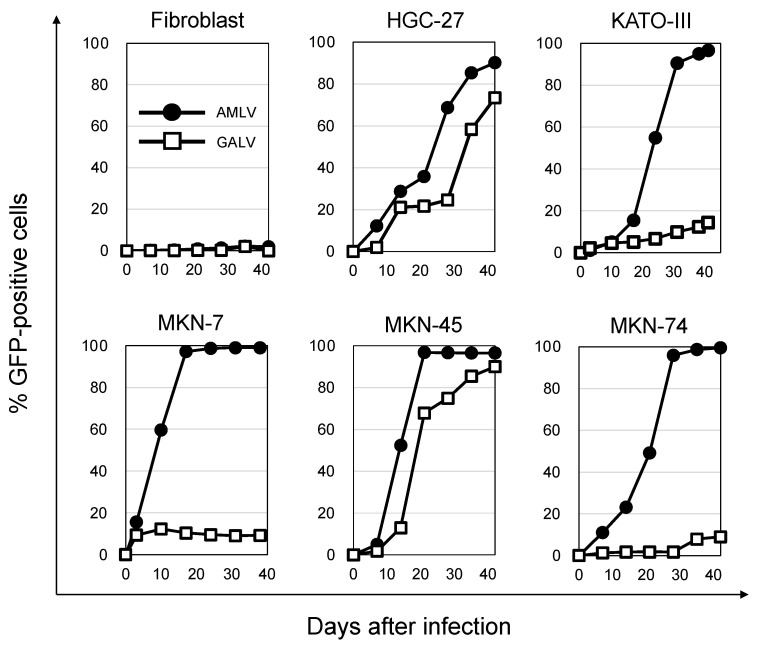
RRV replication and spread in vitro. Human gastric cancer cell lines (HGC-27, KATO-III, MKN-7, MKN-45, MKN-74) and fibroblasts were inoculated with AMLV-GFP (closed circle) or GALV-GFP (open square) vectors (MOI = 0.01). On the days of passage, cells were analyzed for GFP expression using flow cytometry. Data are representative of 3 independent experiments, all yielding similar results.

**Figure 3 ijms-24-14823-f003:**
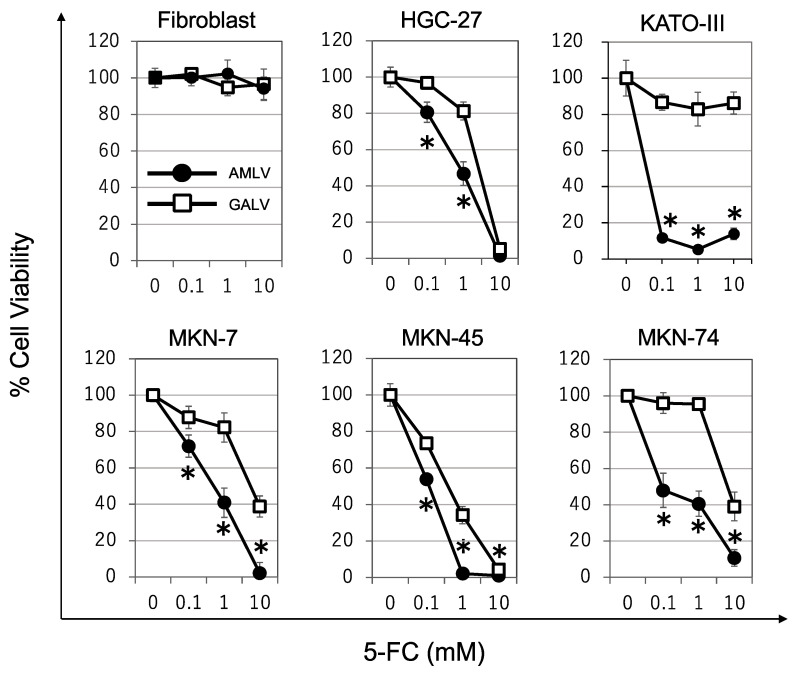
Cell-killing efficiency by RRVs in vitro. Human gastric cancer cell lines (HGC-27, KATO-III, MKN-7, MKN-45, MKN-74) and fibroblasts were infected with AMLV-CD (closed circle) or GALV-CD (open square) (MOI = 0.01). Then, 5-FC was administered to all gastric cancer cells and fibroblasts once, 16 days after RRV infection. Cell viability was examined using AlamarBlue assay 6 days after 5-FC administration. Data represent the mean ± SD (*n* = 3). * *p* < 0.05.

**Figure 4 ijms-24-14823-f004:**
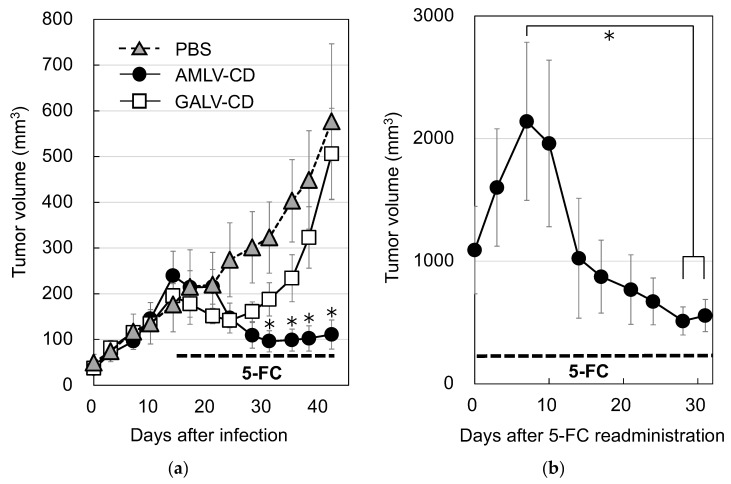
RRV-mediated prodrug activator gene therapy of primary and recurrent subcutaneous MKN-74 human GC xenograft models. (**a**) Antitumor effect of RRV/5-FC on a nude mouse subcutaneous xenograft model of human gastric cancer. To induce the subcutaneous tumor model of human gastric cancer, MKN-74 cells (1 × 10^6^ cells per mouse) were transplanted subcutaneously into nude mice. Tumors were injected with AMLV-CD (closed circle) (*n* = 13), GALV-CD (open square) (*n* = 12), or PBS (gray triangle) (*n* = 12) when the tumor diameter reached 5 mm. The 5-FC was administered to mice three times a week for 35 days. Data are presented as the mean ± SE. * *p* < 0.05. (**b**) Resumption of 5-FC in a recurrent tumor model. MKN-74 recurrent tumors were prepared from mice after 5-FC discontinuation in the AMLV-treated group in Figure 4a (*n* = 6), as described in the Materials and Methods. Then, 5-FC was re-administered three times a week for 31 days. Data are presented as the mean ± SE. * *p* < 0.05.

## Data Availability

The data presented in this study are available on request from the corresponding author.

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
