# Peer review of "Retroviral Replicating Vectors Mediated Prodrug Activator Gene Therapy in a Gastric Cancer Model"

_ijms, 2023, doi:10.3390/ijms241914823_

Round 1
Reviewer 1 Report
This paper described a therapy to kill gastric cancer cells using retroviral replicating vectors that introduce prodrug-activating genes into tumor cells, causing cell death upon subsequent prodrug administration. The study compared two specific retrovial replicating vectors and found that amphotropic murine leukemia virus had significantlly greater antitumor effects on human gastric cancer cells compared to gibbon ape leukemia virus.
The study method is described in detail, and the overall presentation is good. However, the manuscript could benefit from adding a list of all abbreviations in the paper.
Author Response
We would like to sincerely thank the Reviewer for recognizing that “The study method is described in detail, and the overall presentation is good.”.
“However, the manuscript could benefit from adding a list of all abbreviations in the paper.”
This point is well taken, and we have now added a list of abbreviations as a supplemental table, since IJMS does not have a section for such a list.
Reviewer 2 Report
The article explores the promising potential of Retroviral Replicating Vectors (RRVs) as a targeted approach for treating human gastric cancer (GC). RRVs are designed to selectively replicate within tumor cells and introduce prodrug-activating genes, allowing for subsequent prodrug administration to induce cell death specifically within the infected tumor cells. The study assesses two distinct RRVs derived from amphotropic murine leukemia virus (AMLV) and gibbon ape leukemia virus (GALV), both of which utilize different receptors (PiT-2 and PiT-1, respectively) for cell entry.
One significant finding in the study is that human GC cell lines exhibited higher expression levels of PiT-2 compared to PiT-1, which aligns with AMLV's utilization of PiT-2. As a result, AMLV, encoding a green fluorescent protein gene, demonstrated more efficient infection and replication within most GC cell lines in comparison to GALV. This observation of preferential infection of GC cells is crucial for the targeted therapeutic approach.
Another important aspect highlighted in the study is the comparison of AMLV and GALV in terms of their ability to induce cytotoxicity when coupled with a cytosine deaminase prodrug activator gene. AMLV exhibited superior performance in promoting cytotoxicity induced by the prodrug 5-flucytosine (5-FC) compared to GALV. Importantly, this difference in efficacy was correlated with the expression levels of viral receptors and the spread of the virus within the tumor cells.
In preclinical models, specifically the MKN-74 subcutaneous xenograft models, AMLV demonstrated significant anti-tumor effects as compared to GALV. Additionally, in a recurrent tumor model involving MKN-74 cells where 5-FC administration was discontinued temporarily, the resumption of 5-FC treatment resulted in a reduction in tumor volume. This indicates the potential for RRV-mediated prodrug activator gene therapy to not only combat existing tumors but also to manage recurrent tumors effectively.
This study underscores the promise of RRV-mediated prodrug activator gene therapy, with a focus on AMLV, as a potentially beneficial strategy for treating human gastric cancer. The specificity of RRVs for tumor cells, coupled with their ability to enhance prodrug-induced cytotoxicity, presents an exciting avenue for further exploration and potential clinical applications in the management of this challenging malignancy. Future research and clinical trials will be instrumental in assessing the safety and efficacy of this approach in human patients.
Author Response
We appreciate you reading our manuscript and appreciating its significance.
Reviewer 3 Report
In this study, the authors compared transfection efficiency of two different RRVs and their potential to promote 5-FC-induced cytotoxicity. It has been demonstrated that AMV exhibited higher ability to infect human gastric cells and significantly inhibited tumor growth compare to GALV, which pave a way for gene therapy-mediated prodrug strategy. This manuscript clearly describe the experimental procedures and discussed the possible mechanism behind the superior anti-tumor effects. It would be more convincing and therefore qualified to be published if the authors can also provide western blotting results of PiT-2 or PiT-1 expression levels on different cell types.
Author Response
We would like to thank the Reviewer for the assessment that “This manuscript clearly describe the experimental procedures and discussed the possible mechanism behind the superior anti-tumor effects.”.
“It would be more convincing and therefore qualified to be published if the authors can also provide western blotting results of PiT-2 or PiT-1 expression levels on different cell types.”
We certainly agree with the Reviewer that the confidence level would be much higher if the results of western blotting were available. We actually tried western blotting, but unfortunately, we could not obtain reliable results, probably due to the lack of good commercially available antibodies. This will take further time to resolve, but we hope to have it resolved by the time of the next paper.